# Galectin-9 as a Potential Modulator of Lymphocyte Adhesion to Endothelium via Binding to Blood Group H Glycan

**DOI:** 10.3390/biom13081166

**Published:** 2023-07-26

**Authors:** Eugenia M. Rapoport, Ivan M. Ryzhov, Ekaterina V. Slivka, Elena Yu. Korchagina, Inna S. Popova, Sergey V. Khaidukov, Sabine André, Herbert Kaltner, Hans-J. Gabius, Stephen Henry, Nicolai V. Bovin

**Affiliations:** 1Shemyakin and Ovchinnikov Institute of Bioorganic Chemistry RAS, 16/10 Miklukho-Maklaya Str., Moscow 117997, Russia; eugenia_rapoport@mail.ru (E.M.R.); innussik.popova@gmail.com (I.S.P.); khsergey54@mail.ru (S.V.K.); 2Faculty of Veterinary Medicine, Ludwig-Maximilians-University, Veterinär Str. 13, D-80539 Munich, Germanykaltner@lmu.de (H.K.);; 3School of Engineering, Computer and Mathematical Sciences, Faculty of Design and Creative Technologies, Auckland University of Technology, Private Bag 92006, Auckland 1142, New Zealand

**Keywords:** adhesion, ABH glycans, glycolipids, endothelial cells, galectin-9, lymphocytes

## Abstract

The recruitment of leukocytes from blood is one of the most important cellular processes in response to tissue damage and inflammation. This multi-step process includes rolling leukocytes and their adhesion to endothelial cells (EC), culminating in crossing the EC barrier to reach the inflamed tissue. Galectin-8 and galectin-9 expressed on the immune system cells are part of this process and can induce cell adhesion via binding to oligolactosamine glycans. Similarly, these galectins have an order of magnitude higher affinity towards glycans of the ABH blood group system, widely represented on ECs. However, the roles of gal-8 and gal-9 as mediators of adhesion to endothelial ABH antigens are practically unknown. In this work, we investigated whether H antigen–gal-9-mediated adhesion occurred between Jurkat cells (of lymphocytic origin and known to have gal-9) and EA.hy 926 cells (immortalized endothelial cells and known to have blood group H antigen). Baseline experiments showed that Jurkat cells adhered to EA.hy 926 cells; however when these EA.hy 926 cells were defucosylated (despite the unmasking of lactosamine chains), adherence was abolished. Restoration of fucosylation by insertion of synthetic glycolipids in the form of H (type 2) trisaccharide Fucα1-2Galβ1-4GlcNAc restored adhesion. The degree of lymphocyte adhesion to native and the “H-restored” (glycolipid-loaded) EA.hy 926 cells was comparable. If this gal-9/H (type 2) interaction is similar to processes that occur in vivo, this suggests that only the short (trisaccharide) H glycan on ECs is required.

## 1. Introduction

The recruitment of leukocytes to a site of inflammation results from a three-step adhesion cascade [1]. In the first step, leukocytes circulating in blood start to roll on the endothelial surface; this process, known as rolling, is in part mediated by P- and E-selectins on endothelial cells (EC) and L-selectin on leukocytes reacting with specific oligoglycans. Rolling also requires the binding of CD44 [1,2,3,4] to the hyaluronic acid of EC. Both selectin-mediated and CD44-mediated rolling cause the activation of leukocytes. Activated leukocytes then express integrins of the β1 and β2 family, which bind to intercellular cell adhesion molecules ICAM-1-5 and VCAM-1, leading to the inhibition of leukocyte rolling and the facilitation of attachment to the endothelium. Adhesion is a key step of the cascade, since attached leukocytes cross the endothelial barrier and migrate to the site of inflammation [5,6].

Galectins (gal) are β-galactoside-binding lectins defined by a shared consensus amino acid sequence for a carbohydrate recognition domain (CRD) [7]. Currently, 15 proteins of this family have been identified in mammals. Based on the structural organization of CRDs, galectins are divided into proto-, chimera-, and tandem-repeat types. Prototype galectins (-1, -2, -7, -10, -13, -14, and -15) contain a single carbohydrate recognition domain (CRD) and form homodimers. Tandem-repeat galectins (-4, -6, -8, -9, and -12) have two non-identical CRDs linked by a short peptide. To the chimeric type belongs only gal-3, which possesses one C-terminal, CRD, and a proline/glycine-rich *N*-terminal domain, through which it is able to form oligomers. Galectins are revealed in numerous cell and tissue types, where they are found in the cytoplasm, nucleus, on the cell surface, and in the intercellular space. Inside, cytoplasm galectins transmit cellular signals and regulate the cell cycle, while on the cell surface, they mediate cell–cell, and cell–matrix adhesion via binding to glycans. Some galectins are mediators of inflammation, markers of tumor progression, and chemo-attractants.

Along with integrins, galectins (gal) mediate leukocyte adhesion; their ligands are oligolactosamines of EC carried by glycoproteins [7]. In particular, gal-3 induces the migration of eosinophils into the endothelium via binding to oligolactosamines in VCAM-1 and integrins [8,9], as well as mediates cytokine secretion via binding to *N*-glycans of CD146 on EC [10], whereas gal-8 mediates lymphocyte adhesion to the sialylated glycans of podoplanin and ALCAM [11,12].

Diverse ligands are presented on EC, besides the oligolactosamines mentioned above, including ABH blood group system antigens [13,14]. It should be noted that (with the very rare exception of the *hh* genotype [15]) all individuals have ABO blood groups and have H glycans, albeit with much higher levels of H antigen in group O individuals than in those with A and B antigens. Although ABH antigens are well-known as galectin ligands on bacteria [16,17], there is a lack of information on the participation of these glycans on ECs interacting with galectins. In this work, using as a cell model the human Jurkat T-cell line, we investigated whether gal-8 and gal-9 could recognize and use H glycan on EA.hy 926 endothelial cells (HUVEC origin, [18]). These cells express only H glycans, due to the lack of A- and B-glycosyltransferase genes [13].

## 2. Material and Methods

### 2.1. Reagents

DMEM-F12 and RPMI-1640 media, glutamine, and goat anti-rabbit IgG conjugated with Alexa Fluor 594 (IgG-Alexa594) were from Invitrogen Co. (Carlsbad, CA, USA). Fetal calf serum (FCS), carbohydrate-free bovine serum albumin (BSA), FITC conjugates of streptavidin (Str-FITC), anti-rabbit IgG (IgG-FITC), and Mowiol^®^ 4-88 were from Sigma-Aldrich (St-Louis, MN, USA). Bovine kidney α1-2fucosidase (20,000 U/mL) was from Biolabs (Ipswich, MA, USA). Biotinylated *Ulex europaeus* agglutinin (UEA I, which reacts with Fucα1-2Galβ1–4GlcNAc-terminating glycans [19]) was from Vector Laboratories (Burlingame, CA, USA). Immuno-Brite Fluorospheres (10 μm) containing 31,000, 115,000, or 450,000 fluorescein residues were obtained from Beckman Coulter Life Sciences (Indianapolis, IN, USA). The fluorescent dye 1,6-diphenylhexatriene (DPH) was from Serva (Heidelberg, Germany).

The synthetic glycolipid construct (Function-Spaсer-Lipid) FSL-H (type 2) (Figure 1) was synthesized as described [20] and could be obtained from GlycoNZ, Auckland, NZ (0089-FSL). Briefly, aminopropylglycoside of H (type 2) trisaccharide (0.010 mmol, 5.9 mg) was dissolved in DMSO (0.5 mL), and this solution was added in 5 equal portions (100 μL) every 10 min to a solution of disuccinimide of adipic acid Ad(ONSu)_2_ (34 mg, 0.10 mmol) in 0.5 mL of DMSO. After the last portion was added, the mixture was kept for 20 min at room temperature and subjected to gel filtration (Sephadex LH-20, elution with CH_3_CN–H_2_O, 0.3% AcOH), followed by freeze-drying from an aqueous solution of AcOH (0.3%) to turn 7.8 mg (95%) of activated *N*-succinimide ester of H (type 2) trisaccharide into a white foam, *R_f_* 0.46 (CHCl_3_–EtOH–H_2_O, 4:9:2, 0.1% AcOH). The obtained *N*-succinimide ester of H (type 2) trisaccharide (0.0043 mmol, 3.5 mg) was dissolved in DMF (300 μL), and the solution was added in 3 equal portions (100 μL every 1.5 h) to the solution of amine H_2_N-CMG-DOPE (5.3 mg, 0.0029 mmol) in the mixture of NaHCO_3_ aq (50 mM, 0.5 mL) and *i*-PrOH (0.25 mL) (pH 8.5). The mixture was left overnight, then neutralized with AcOH (2 μL), and then subjected to gel filtration (Sephadex LH-20, elution with *i*-PrOH–H_2_O, 1:2, 0.25% AcOH, 0.5% Py). Fractions containing pure product were freeze-dried from water twice, and the obtained residue was dissolved in H_2_O (1 mL) and titrated with NaHCO_3_ aq (50 mM) to pH 6.7 to convert the product to Na_5_-salt. The resulting solution was freeze-dried to obtain 6.2 mg (82%) of the CMG-FSL construct of H (type 2) trisaccharide as a white foam; *R*_f_ 0.37 (*i*-PrOH–MeOH–CH_3_CN–H_2_O, 4:3:6:4); ^1^H NMR (D_2_O–CD_3_OD, 2:1, characteristic signals): *δ* 5.40–5.34 (m, 4H, 2 -C*H*=C*H*-), 5.32 (d, 1H, *J*_1,2_ 3.2, H-1^Fuc^), 5.31–5.27 (m, 1H, OCH_2_C*H*OCH_2_O-), 5.04 (q, 1H, *J*_5,6_ 6.6, H-5^III^), 4.55 (d, 1H, *J*_1,2_ 7.7, H-1^Gal^), 4.50 (d, 1H, *J*_1,2_ 8.4, H-1^GlcNAc^), 4.46 (dd, 1H, *J* 2.4, *J* 12.3, -C(O)OC*H*HCHOCH_2_O-), 2.43–2.24 (m, 12H, 6 -C*H*_2_CO), 2.08–2.00 (m, 11H, 2 -C*H*_2_-CH=CH-C*H*_2_- and NHC(O)C*H*_3_), 1.82–1.75 (m, 2H, OCH_2_C*H*_2_CH_2_NH), 1.69–1.57 (m, 12H, 2 COCH_2_C*H*_2_C*H*_2_CH_2_CO and 2 COCH_2_C*H*_2_-), 1.41–1.26 (m, 40H, 20 C*H*_2_), 1.25 (d, 3H, *J*_5,6_ 6.6, H-6^Fuc^), 0.91 (t, *J* 7.1, 6H; 2 C*H*_3_).

### 2.2. Galectins and Galectin-Specific Antibodies

Recombinant human gal-8 and gal-9 were prepared as previously described [21] and purified by affinity chromatography on lactosylated Sepharose 4B. Polyclonal antibodies against these recombinant galectins were raised in rabbits. The IgG fraction was isolated by affinity chromatography using protein-A Sepharose 4B (Pharmacia, Freiburg, Germany) and checked for a lack of cross-reactivity against other lectin family members using Western blotting and ELISA. Cross-reactivity, if present, was removed by affinity chromatography on galectin-specific resins [22].

For studies of Jurkat cell adhesion to EA.hy 926 cells, polyclonal antibodies against endogenic gal-9 obtained from Antibody Verify Co (Las Vegas, CA, USA) were used.

### 2.3. Cell Culture

Endothelial EA.hy 926 cells, (kindly provided by Dr. C.-J. Edgell, University of North Carolina, Chapel Hill, NC, USA) and Jurkat cells (human T-lymphoblasts originating from acute T cell leukemia, clone E6-1, ATCC^®^ TIB-152) were cultured in DMEM-F12 or RPMI-1640 media, respectively, supplemented with 10% FCS and 2 mM glutamine at 37 °C in a humidified atmosphere of 5% CO_2_.

### 2.4. Enzymatic Defucosylation of EA.hy 926

Cells (1 × 10^6^ cells/mL) were grown to confluence in medium using 24-well plates (Nunc, Roskilde, Denmark), washed three times with DMEM-F12-0.3% FCS, and treated with fucosidase (4 U/mL) at 37 °C in a humidified atmosphere of 5% CO_2_ overnight. The next day, cells were detached with Versene solution (PBS containing 0.02% EDTA *v*/*v*) and washed three times with phosphate-buffered saline containing 0.2% BSA (PBA) using centrifugation at 450× *g* (Jouan, rotor T20, France) at 4 °C. The binding of labeled UEA I served as a control for defucosylation. Briefly, the cells (1 × 10^5^ cells per well in 50 μL) were carefully resuspended in PBA and incubated with 50 μL of biotinylated UEA I in PBA (final concentration 20 μg/mL) for 30 min at 4 °C under gentle agitation on a shaker, followed by incubation with FITC-labeled streptavidin (1:50 dilution in PBA) for 20 min under the same conditions. The cells were washed three times with PBA and analyzed by flow cytometry.

### 2.5. Insertion of FSL-H (Type 2) into EA.hy 926 Cells

Cells (1 × 10^6^ cells/mL) were grown to confluence in medium using 24-well plates (Nunc, Denmark), washed three times with DMEM-F12-0.3% FCS, treated with fucosidase as described above, washed three times with PBA, and incubated with FSL-H (type 2) (final concentration: 5 μM in PBA) at 37 °C in a humidified atmosphere of 5% CO_2_ for 1 h. FCS was used in a low concentration (0.3% FCS in DMEM-F12 in medium), as high percentages of serum can interfere with glycolipid insertion into cell membranes [23,24]. To analyze FSL-H (type 2) insertion, cells were detached with Versene solution, washed three times with PBA by centrifugation at 95× *g* and 4 °C for 3 min, and incubated with biotinylated UEA I for 30 min at 4 °C under gentle agitation on a shaker. Cells were then incubated with Str-FITC (1:50 dilution in PBA) for 20 min under the same conditions. The cells were washed three times with PBA and analyzed by flow cytometry.

### 2.6. Binding of Galectins to EA.hy 926 Cells

After the insertion of FSL-H (type 2), cells were detached with Versene solution and washed three times by centrifugation, as described above. Aliquots of the cell suspension (1 × 10^5^ cells in 50 μL) were incubated with 50 μL of gal-8 or gal-9 (0.1 mg/mL) for 30 min at 4 °C under gentle agitation on a shaker. To remove unbound galectins, cells were carefully washed once using centrifugation under the previously described conditions [25]. The measurement of galectin binding was determined with galectin-specific biotinylated antibodies (5 μg/mL in PBA) and Str-FITC (1:50 dilution in PBA) at 4 °C for 20 min. As a negative control, native galectin-free cells were incubated with the galectin-specific antibody.

### 2.7. Flow Cytometry

After the washing steps, cells were transferred into a tube and mixed with 2 mL of PBS. Flow cytometry was performed at room temperature using a FACScan instrument (Becton-Dickinson Co, Franklin Lakes, NJ, USA) equipped with the software FlowJo V10.5.3 or a FC500 cytofluorimeter (Beckman Coulter, Miami, FL, USA) equipped with the software Kaluza 1.3. Live cell populations were first gated using morphological parameters of forward light scatter (FSC) vs. sideward light scatter (SSC). Then, the logarithmic fluorescence intensity (FL-FITC) of the gated population was measured. The fluorescence (in some articles, this parameter is called *Fluorescence increase*) shown in the figures was calculated as [(F_i_/F_0_) × 100] − 100, where F_i_ is the geometric mean of the fluorescence intensities of cells after incubation with anti-galectin + second IgG-FITC or UEA I + Str-FITC, and F_0_ is the geometric mean of the fluorescence intensities of cells stained only with IgG-FITC or Str-FITC. In parallel, the fluorescences of microspheres (in the dilution range recommended by the manufacturer) were measured. The values of the obtained fluorescence intensities were used to calculate the number of FSL-H (type 2) molecules per cell.

### 2.8. Detection of FSL-H (Type 2) in the Cell Membrane

To visualize the membrane, DPH stock solution (8 mM) in dimethyl sulfoxide was prepared and added to the cells (final concentration: 4 μM). Cells were incubated with DPH at 4 °C for 1 h and then washed with PBS. The inserted FSL-H (type 2) was detected with biotinylated UEA I followed by Str-FITC, as described above. Microscopy mounting solution containing 2.4 g of Mowiol 4-88, 6 g of glycerol, 6 mL of water, and 12 mL of 0.2 М Tris-HCl (рH 8.5) was placed on the microscope slide, followed by the cell suspension (10 μL). All images were obtained with a confocal microscope Nikon Eclipse TE-2000-E (Nikon, Minato City, Japan) and analyzed with ImageJ. At least ten randomly selected cells were analyzed in each experiment.

### 2.9. Cell Adhesion Assay

The adhesion of Jurkat cells to EA.hy 926 cells was evaluated using a confocal microscope (Nikon Eclipse TE-2000-E). Briefly, EA.hy 926 cells (1 × 10^6^ cells/mL) were grown to confluence in DMEM-F12 medium, washed with DMEM-F12 medium containing 0.3% FCS, and treated with fucosidase, as described above. FSL-H constructs were then inserted into the EA.hy 926 cells, as described above, cells were washed with RPMI-0.3% FCS, and then Jurkat cells (1 × 10^6^ cells/well) in the same media were added. The plate was incubated at 37 °C in a humidified atmosphere of 5% CO_2_ overnight. The next day, non-adhered cells were removed by gentle washing with РBA. Jurkat cells adhered to the monolayer of EA.hy 926 were stained with endogenic anti-gal-9 and IgG-Alexa Fluor 594. Three confocal microscope images per well were taken, and adherent cells were counted using ImageJ software 1.53t.

To confirm involvement of gal-9 in adhesion inhibition of its binding to EA.hy 926 cells by anti-gal-9 antibodies and polyacrylamide conjugate H (type 2)-PAA was performed. Jurkat cells were incubated with antibodies against gal-9 or H (type 2)-PAA (100 µM by trisaccharide) for 60 min at 37 °C and then added to EA.hy 926 cells.

### 2.10. Statistical Analysis

Data represent means +/− standard deviations. The unpaired Student’s *t*-test for statistical analysis of the results was used.

## 3. Results

### 3.1. Binding of Galectins to Unmodified and Fucosidase-Treated EA.hy 926 Cells

To determine if gal-8 and gal-9 recognized cell surface-exposed H glycans, the binding of galectins to unmodified cells was compared against defucosylated cells (i.e., treated with fucosidase). Quantitative data were obtained using flow cytometry, measuring the fluorescence intensity’s (MFI) geometric mean. The decreased binding of UEA I (a plant lectin recognizing the Fucα1-2Galβ1-4GlcNAc glycotope [19]) verified the level of defucosylation. (The fluorogram is presented in Appendix A.)

After treatment with fucosidase, gal-9 binding to EA.hy 926 cells significantly decreased (Figure 2B), while defucosylation did not affect gal-8 binding (Figure 2A). The concentrations of 7 μg/mL of gal-8 and 55 μg/mL of gal-9 were found to be required to reach 50% binding, indicating that gal-9 binds predominantly to H-glycans, while gal-8 utilizes other glycans to attach to cells.

Because only gal-9 could bind the H antigen of EA.hy 926, further experiments were conducted solely with gal-9.

### 3.2. Binding of Gal-9 to Defucosylated EA.hy 926 Cells after Their “Refucosylation” with FSL-H (Type 2)

FSL-H (type 2) (Figure 3 and Appendix A) was inserted in defucosylated cells, and UEA I binding to these defucosylated cells increased four-fold following FSL-H (type 2) insertion (see in Appendix A), indicating a significant degree of modification. Comparing the fluorescence of FSL-H (type 2)-modified cells with Immuno-Brite fluorosphere-calibrating particles, it was calculated that the number of FSL-H (type 2) molecules inserted per cell was ~10^5^ (after 1 h of incubation).

The binding of gal-9 to FSL-H (type 2)-modified cells was determined, and fluorograms are presented in Appendix A. The insertion of FSL-H (type 2) into defucosylated cells resulted in a three-fold increase in gal-9 and at a level that was comparable to unmodified cells (Figure 3). The low signal seen in the defucosylated cells indicated incomplete defucosylation.

### 3.3. Localization of Inserted FSL-H (Type 2)

The FSL-H (type 2) distribution in the EA.hy 926 cell membrane was assessed using confocal microscopy. For membrane tracing, the fluorescent dye DPH (1,6-diphenylhexatriene), which has no polar head and is known to accumulate and fluoresce only in a hydrophobic environment [26], was used. FSL-H (type 2) stained with UEA I/Str-FITC was detected near the DPH staining zone; its distribution was similar to that of H glycan in unmodified cells (Figure 4). The fucosidase-treated cells showed poor staining with UEA I, which reflected, as mentioned above (see in Appendix A), the expected incomplete defucosylation. These results showed that FSL-H (type 2) was able to restore H reactivity to a level similar to that seen on unmodified cells, and that defucosylated cells only had trace levels of UEA I-reactive H glycan.

### 3.4. Gal-9 Promotes Adhesion of Jurkat Cells to EA.hy 926 Cells

The adhesion of Jurkat cells to EA.hy 926 cells was assessed. Jurkat cells were added to a monolayer of EA.hy 926 cells, with variable degrees of FSL-H modification. The adhesion of Jurkat cells (as a measure of galectin-9 on the cell surface) to unmodified cells, fucosidase-treated cells, and FSL-H (type 2) fucosylation-restored cells were compared. Using the number of Jurkat cells attached to unmodified EA.hy 926 cells as 100%, calculations revealed that only 4% of Jurkat cells attached to defucosylated EA.hy 926 cells (Figure 5A,B,D), whereas 50% of Jurkat cells attached to defucosylated EA.hy 926 cells after the insertion of FSL-H (type 2) (Figure 5C,D).

To confirm that gal-9 mediated the adhesion of Jurkat to EA.hy 926 cells, we evaluated adhesion assay in the presence of anti-gal-9 antibodies or H (type 2)-PAA [27]. (Previously, we used a similar assay for the study of galectin specificity [27,28].) Both reagents inhibited Jurkat cell adhesion (Figure 6). We explained the weak inhibition in the case of anti-gal-9 with the fact that only a small population of these polyclonal antibodies were directed to the carbohydrate recognition domain of gal-9.

## 4. Discussion

The interaction of β1 and β2 integrins of leukocytes with intercellular adhesion molecules of the ICAM and VCAM families [5,6] on activated EC mediated the firm adhesion of leukocytes to the endothelium. Other adhesion mechanisms of this type were also known, including leukocyte CD44 reacting with endothelial hyaluronic acid [2,3,29]. There was also data on galectin-mediated adhesion of leukocytes, which recognize oligolactosamines on the endothelium [7] and other ligands, including glycans of the ABH blood group system (which are widely represented on EC glycolipids and glycoproteins) [13,14,30]. However, there was no information about galectin involvement in ABH-mediated adhesion to endothelium, although the affinity of tandem-repeat-type galectins to ABH glycans was reported [31,32,33]. It was an order of magnitude higher than oligolactosamines [27,34]. The extracellular gal-8 and gal-9 were present in the glycocalyx of leukocytes [7]. As mentioned above, tandem-repeat-type galectins had two homologous carbohydrate recognition domains (CRD), namely N-CRD and C-CRD [34]. N-CRD exhibited a higher affinity for oligolactosamines [27], while C-CRD “targeted” ABH antigens. In this work, we examined whether gal-8 and gal-9 were able to mediate the adhesion of lymphocytes to EC. To achieve this, we used a model system, namely, the interaction of galectin-positive Jurkat cells (T-lymphocyte origin [35] with EA.hy 926 endothelial cells (i) bearing natural H antigen [13], (ii) being artificially depleted of H antigen by fucosidase treatment, and (iii) defucosylated then refucosylated with only H (type 2) antigen (using synthetic FSL glycolipid)). Due to the absence of ABO glycosyltransferases in EA.hy 926 cells, only the involvement of H (type 2) was examined. Additionally, the lack of gal-8 and gal-9 in the glycocalyx of EA.hy 926 cells (unpublished observations) made these cells particularly suitable for this study. It should be noted that although defucosylation was incomplete, the inability of glycosidases to fully deplete glycan-substrates on viable cells was well established [36]. However, despite the residual levels of H glycan remaining, fucosidase-treated cells showed high levels of experimental differentiation from untreated cells. The depletion of fucose residues affected the binding of exogenic gal-9 but not gal-8, indicating that fucose was not critical for gal-8 binding. It is known that the affinity of gal-8 to H glycan was an order of magnitude lower than that of gal-9 [27], and the most potent fucose-containing glycan for gal-8 interaction was blood group A tetrasaccharide, which was absent on EA.hy 926 cells. We believe that on EA.hy 926 cells, gal-8 bound to oligolactosamines, the amount of which increased significantly due to the defucosylation of native glycans. On the contrary, H glycans (type 1 and type 2) were the most potent ligands for gal-9, and their affinities were found to be much stronger compared to oligolactosamines and α2-3-sialylated glycans [27,28].

Fucosidase treatment disrupted H glycotopes and a large range of other fucose-related histo-blood group antigens. Therefore, to confirm the observations that the loss of gal-9 activity in defucosylated cells was due to H glycans, we restored fucosylation in cells with a defined blood group H (type 2) antigen. This was performed by using a synthetic glycolipid FSL-H (type 2) and the well-established principles and methodologies of Kode Technology [37]. As a result of refucosylation with H (type 2), the ability of gal-9 to bind EA.hy 926 cells was restored

Thus, the presence of the trisaccharide H (type 2) was sufficient for the interaction of Jurkat cell gal-9 to EA.hy 926 cells, suggesting that gal-9 of leukocytes was potentially able to mediate or participate in the adhesion via binding to H glycan on EC. Furthermore, this could be extrapolated to suggest that EC cells of different ABO blood types, where the amount of H antigen is much more significant in blood group O than in A and B phenotypes, could have some impact on their immunological response(s).

Gals-1, -3, -8, and -9 were revealed on blood cells [38,39]. Among them, only gal-9 was revealed in cytoplasm and on the surface of Jurkat cells [35,38], where it accumulated as patches [40] in the depth of the glycocalyx. In this study, we showed that gal-9 mediated the adhesion of Jurkat cells to endothelial cells by binding to H-glycans. However, this in vitro data did not necessarily mean that leukocytes also adhered to EC in vivo, due to gal-9/H type 2 interactions, but it suggested the possibility of such a scenario.

## Figures and Tables

**Figure 1 biomolecules-13-01166-f001:**
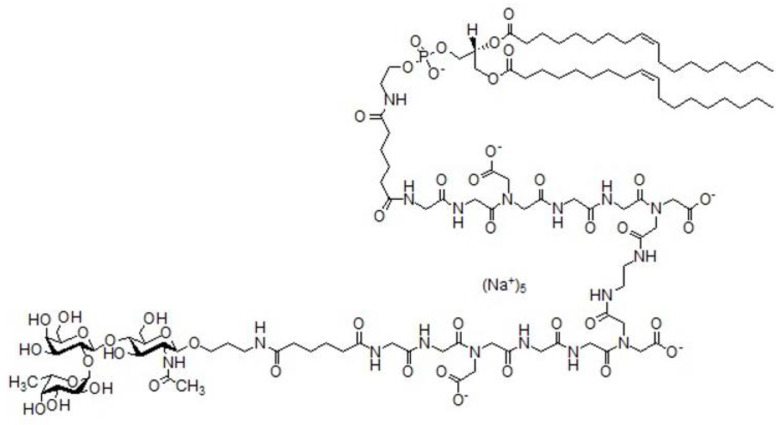
Structure of the synthetic glycolipid analog, FSL-H (type 2). Soluble 30 kDa polyacrylamide glycoconjugate H (type 2)-PAA containing 20% mol of glycan was obtained from Lectinity (Moscow, Russia). All the other reagents were from AO Reachem LLC (Moscow, Russia).

**Figure 2 biomolecules-13-01166-f002:**
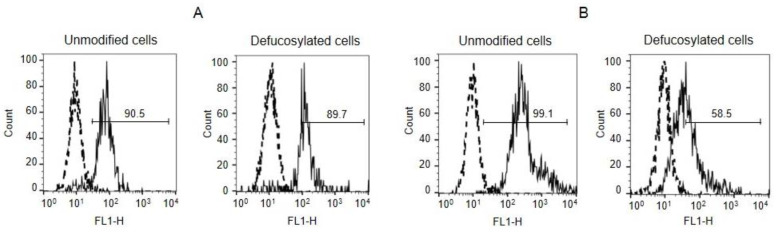
Effect of fucosidase treatment on EA.hy 926 cell-binding with gal-8 (**A**) and gal-9 (**B**), as determined by flow cytometry. Cells were treated with fucosidase, incubated with galectins, and stained with anti-galectin, followed by IgG-FITC, as described in Material and Methods. Results shown include fluorograms (the log of fluorescence intensity FL-1, x-axis, was plotted against the cell number, y-axis), and the number given for the black curve represents the percentage of cells reactive with the lectins. Cells stained with only IgG-FITC were used as a negative (dashed line) control.

**Figure 3 biomolecules-13-01166-f003:**
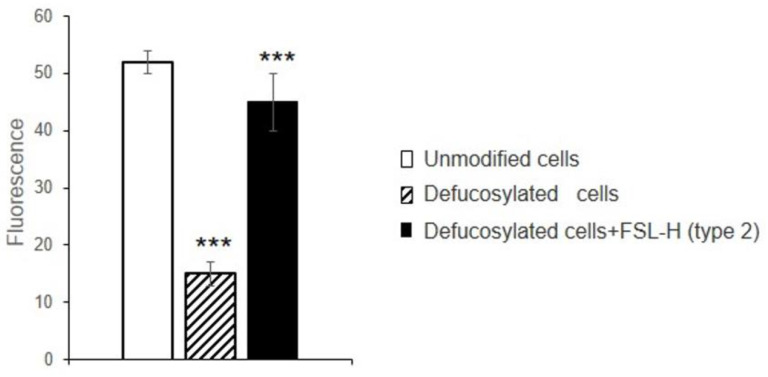
Binding of gal-9 to EAhy 926 cells modified with FSL-H (type 2), as determined by flow cytometry. Cells were treated with fucosidase and incubated with FSL-H (type 2) at 37 °C for 1 h. After insertion and washing, cells were incubated with gal-9 and stained with anti-gal-9, as described in Material and Methods. Data represent the means of fluorescence +/− standard deviations (*n* = 3); *** *p* < 0.001.

**Figure 4 biomolecules-13-01166-f004:**
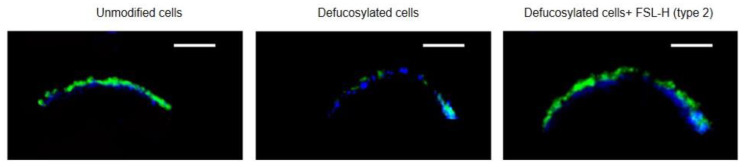
Localization of FSL-H (type 2) in the EA.hy 926 cell membrane, as determined by confocal microscopy. FSL-H (type 2) was inserted in the defucosylated cells, as described in Material and Methods; it was washed with PBS, and the membrane was stained with DPH (4 μM, blue) for 1 h at 4 °C. FSL-H was visualized (in green) with biotinylated UEA I/Str-FITC. The white bar inset at the top right in each image corresponds to 5 μm.

**Figure 5 biomolecules-13-01166-f005:**
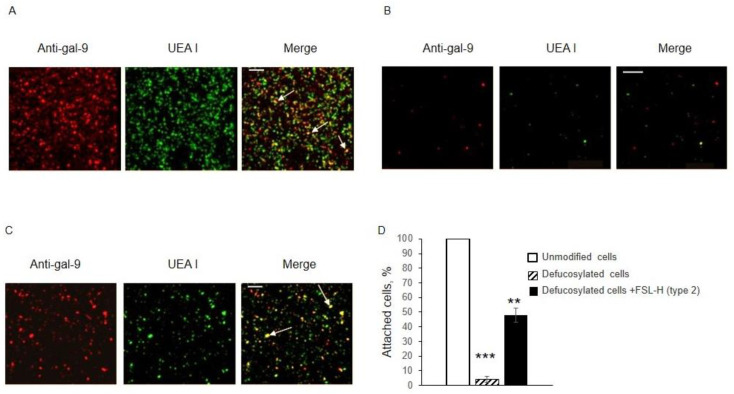
Confocal microscopy visualized the adhesion of Jurkat cells to (**A**) unmodified EA.hy 926 cells, (**B**) defucosylated EA.hy 926 cells, and (**C**) FSL-H (type 2) fucosylation-restored EA.hy 926 cells. Gal-9 was stained red with anti-gal-9 + IgG-Alexa594, and FSL-H (type 2) was stained green with biotinylated UEA I + Str-FITC. Jurkat cells, when adhered to EA.hy 926 and when images were merged, resulted in a yellow color (generated by overlapping red and green fluorescence signals), and examples are indicated with arrows. (**D**) Adherent cells were counted in each well in triplicate and expressed as a percentage of unmodified cell binding +/− standard deviations (*n* = 3); *** *p* < 0.001, ** *p* < 0.01. Bar: 200 µm.

**Figure 6 biomolecules-13-01166-f006:**
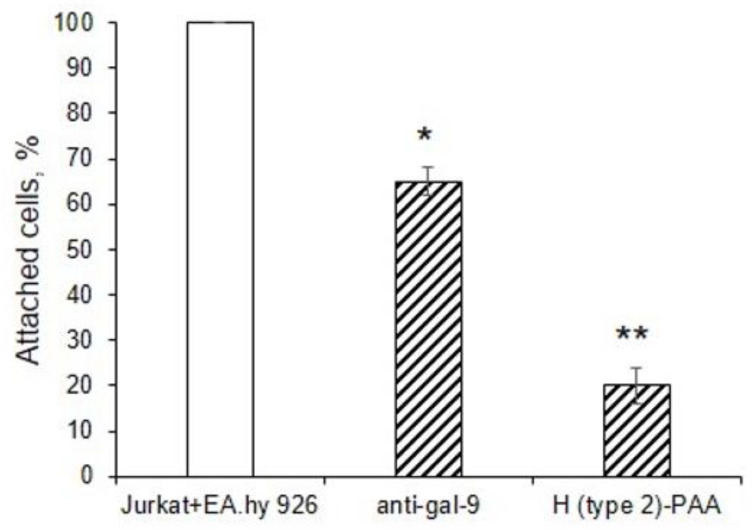
Adhesion of Jurkat cells to EA.hy 926 cells in the presence of inhibitors, shown with confocal microscopy data. Jurkat cells were incubated with anti-gal-9 (10 μg/mL) or H (type 2)-PAA (100 μM) for 1 h at 37 °C, washed to remove the unbound inhibitor, and added to the EA.hy 926 cells. The adhesion cell assay was performed, as described in Material and Methods. To visualize the attached Jurkat cells, DAPI reagent (4′,6-diamidino-2-phenylindole) was used. Adherent cells were counted in each well in triplicate. Data represent percentage of Jurkat cells bound to EA.hy 926 cells +/− standard deviations (*n* = 3); ** *p* < 0.01, * *p* < 0.1.

## Data Availability

Not applicable.

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
