# Peer review of "Galectin-9 as a Potential Modulator of Lymphocyte Adhesion to Endothelium via Binding to Blood Group H Glycan"

_biomolecules, 2023, doi:10.3390/biom13081166_

Round 1

Reviewer 1 Report

The manuscript by Rapoport et al gives insight into the role of galectin-9 in the association of Jurkat and endothelial cells and hypothesizes about the role of fucosylation of H-antigen in this process. The manuscript is well written and the conclutions are interesting. However, several points need to be addressed before it is accepted for publication.

1. The authors should add a more specific quantification of the binding of Gal-9, and -8 to H-antigen (fucosylated or non-fucosylated) by determining IC50 or Kd. Recombinant galectins may be used for this aim. Thus, conclusions of the manuscript may be better discussed.

2. The authors should asssess the binding between the cells without the presence of Gal-9 (Gal-8) - e.g. by knock-out, or at least by competitive inhibition.

3. The authors should add a western blot showing the expression of major galectins (-1 and -3) and of other galectins relevant for cell adhesion on the used cell lines. As known, galectin expression may also change with cell passege, etc. How do you prove that it is in in particular Gal-9 that mediates this adhesion process and not other galectins?

Minor points:

Supplementary Figures should contain legends directly at the figures.

Please specify the fucosidase used in the experiments.

The introduction should contain a section on the structure of galectins and their role in cell adhesion.

Author Response

Please see our answer as attached file 

Reviewer 2 Report

The authors of the manuscript titled, “Galectin-9 modulates adhesion of lymphocytes to endothelium via binding to blood group H Glycan” present data suggesting that T cell leukemia, Jurkat, cells binding to EA.hy 926 endothelial cells (ECs).  As the title suggests, the authors do not conduct experiments with lymphocytes.  They were conducted with human T cell leukemia Jurkat cells. The authors insinuate that based on 1.) the ability of recombinant galectin-9 (Gal-9) to bind ECs, including to α1,2 fucosidase-treated ECs (but to a lesser degree), and 2.) the purported expression of Gal-9 in Jurkat cells, Jurkat cell adhesion to ECs was due to Gal-9.  The adhesion experiments do not demonstrate Gal-9-mediated adhesion of Jurkat cells to ECs.  Furthermore, the binding experiments do not demonstrate that Gal-9 binds directly to H-antigen.  While removal of α1,2 fucosylation on Fucα1,2Galß1,4GlcNAc-R H-antigen moiety with α1,2 fucosidase or addition of H-antigen on the surface of cells did alter overnight adhesion of Jurkat cells to ECs, these conditions do not infer that Gal-9 is binding to H-antigen. Moreover, the adhesion of Jurkat cells to ECs could be due to a number adhesion receptor/ligand molecules, namely integrin/integrin ligands, proteoglycans, selectin/ligands.  This paper is not acceptable for publication and should be rejected.

The authors of the manuscript titled, “Galectin-9 modulates adhesion of lymphocytes to endothelium via binding to blood group H Glycan” present data suggesting that T cell leukemia, Jurkat, cells binding to EA.hy 926 endothelial cells (ECs).  As the title suggests, the authors do not conduct experiments with lymphocytes.  They were conducted with human T cell leukemia Jurkat cells. The authors insinuate that based on 1.) the ability of recombinant galectin-9 (Gal-9) to bind ECs, including to α1,2 fucosidase-treated ECs (but to a lesser degree), and 2.) the purported expression of Gal-9 in Jurkat cells, Jurkat cell adhesion to ECs was due to Gal-9.  The adhesion experiments do not demonstrate Gal-9-mediated adhesion of Jurkat cells to ECs.  Furthermore, the binding experiments do not demonstrate that Gal-9 binds directly to H-antigen.  While removal of α1,2 fucosylation on Fucα1,2Galß1,4GlcNAc-R H-antigen moiety with α1,2 fucosidase or addition of H-antigen on the surface of cells did alter overnight adhesion of Jurkat cells to ECs, these conditions do not infer that Gal-9 is binding to H-antigen. Moreover, the adhesion of Jurkat cells to ECs could be due to a number adhesion receptor/ligand molecules, namely integrin/integrin ligands, proteoglycans, selectin/ligands.  This paper is not acceptable for publication and should be rejected.

Author Response

Please see our answer as attached file

Round 2

Reviewer 1 Report

The authors have sufficiently addressed my comments, performing additional experiments and rewording & completing of the manuscript. The manuscript has improved significantly. I recommend the manuscript for acceptance in its present form.

Author Response

We sincerely wish to thank the Reviewer #1 for the comments guiding us to address weak spots in our original version. 

Reviewer 2 Report

1. Supplemental Fig 3 should be moved to primary figures/data.

2. Sentence structure/punctuation/grammar still need some editing.

1. Supplemental Fig 3 should be moved to primary figures/data.

2. Sentence structure/punctuation/grammar still need some editing.

Author Response

Thank you for your letter from July, 18, 2023, with comments on our manuscript ID: biomolecules-2468335.

1) Ref. remarks:  Supplemental Fig 3 should be moved to primary figures/data.

We added the inhibitory assay to the main text of manuscript, as Fig. 6.

2) All the other remarks were taken into account.